# Evaluation of the Quality of Irrigation Machinery by Monitoring Changes in the Coefficients of Uniformity and Non-Uniformity of Irrigation

**Ján Jobbágy** [1] , **Peter Dančanin** [1], **Koloman Krištof** [1,*] , **Juraj Maga** [1] **and Vlastimil Slaný** [2]

[1] Department of Machines and Production Biosystems, Faculty of Engineering,
Slovak University of Agriculture in Nitra, Hlinku 2, 94976 Nitra, Slovakia; jan.jobbagy@uniag.sk (J.J.);
xdancanin@uniag.sk (P.D.); juraj.maga@uniag.sk (J.M.)

[2] Department of Agricultural, Food and Environmental Engineering, Faculty of AgriSciences,
Mendel University in Brno, Zemědělská 1, 61300 Brno, Czech Republic; vlastimil.slany@mendelu.cz

[*] Correspondence: koloman.kristof@uniag.sk

**Abstract:** Recently, the development of agricultural technology has been focused on achieving higher reliability and quality of work. The aim of the presented paper was to examine the possibilities of evaluating the quality of work of wide-area irrigation machinery by monitoring the coefficients of uniformity and non-uniformity of irrigation. The object of the research was pivot irrigation machinery equipped with sprinklers with a total length from 230 to 540 m. The commonly applied quality of work parameter for wide-range irrigators is the irrigation uniformity coefficient according to Heermann and Hein $CUH$. Work quality evaluations were also carried out through other parameters applicable in practice, such as irrigation uniformity coefficients calculated according to Christiansen $CU$, Wilcox and Swailes $C_{ws}$, and our introduced parameters, the coefficient $a_r$ (derived from the degree of unevenness according to Oehler) and the degree of uniformity $\gamma_r$ (derived from the degree of non-uniformity according to Voight). Other applied parameters for determining the quality of work of wide-range irrigation machinery were the coefficients of irrigation uniformity according to Hart and Reynolds $CU_{hr}$, further according to Criddle $CU_{cr}$ and subsequently according to Beale and Howell $CU_{br}$. Next, the parameters of the non-uniformity coefficient according to Oehler $a$, the coefficient of variation according to Stefanelli $C_v$, the degree of non-uniformity according to Voigt $\gamma$ and the degree of non-uniformity according to Hofmeister $E_f$ were evaluated. Field tests were performed during the growing season of cultivated crops (potatoes, corn and sugar beet) in the village of Trakovice (agricultural enterprise SLOV-MART, southwest of the Slovakia) and in the district of Piešťany (Agrobiop, joint stock company). During the research, the inlet operating parameters (speed stage, inlet pressure, irrigation dose), technical parameters (number of sprayers, total length, number of chassis) and weather conditions (wind speed and temperature) were recorded. The obtained results were examined by one-way ANOVA analysis depending on the observed coefficient or input conditions and subsequently verified by Tukey and Duncan tests as needed. Irrigation uniformity values ranged from 67.58% ($C_{ws}$) to 95.88% ($CU_{bh}$) depending on the input conditions. Irrigation non-uniformity values ranged from 8.58 ($a$, $E_f$) to 32.42% ($C_v$). The results indicate a statistically significant effect of the site of interest and thus the impact of particular field conditions ($p < 0.05$). When evaluating the application of different coefficients of irrigation uniformity, the results showed a statistically significant effect only in the first test ($p = 0.03$, $p < 0.05$). During further repeated measurements, the quality of work increased due to the performed inspection of all sprayers and the reduction in the influence of the wind.

**Keywords:** irrigation; coefficients; water; sprinklers; efficiency; quality

## 1. Introduction

In recent years, modern irrigation technologies have come to the forefront of agricultural development for water conservation [1–3]. To achieve high yields and yield stability of the irrigated crop, it is necessary to achieve a high value of the quality of work of irrigators. In addition, [4] has reported a significant reduction in crop yield values due to the decrease in irrigation water at all uniformity levels. One of the possibilities to increase the quality of work is reliable irrigation technology and reliable work of the irrigator [5–7]. Irrigation quality is assessed by the correct intensity and uniformity of irrigation. Intensity expresses the amount of water in millimeters delivered by the sprinkler per unit time [8,9]. Furthermore, [10] concluded that the structural difference of the sprinklers strongly affected the water distribution characteristics, thereby affecting their irrigation performance in the field. The quality of the work of irrigation technology is often evaluated based on the value of the coefficient of irrigation uniformity obtained by calculation from the measured values of the irrigation water in precipitation measuring vessels [11].

One of the first researchers to assess irrigation uniformity was Staebner [12], who followed the rule that maximum irrigation intensity should not exceed twice the minimum, except for the edge zone, when assessing irrigation uniformity. Based on the measurement of precipitation heights in precipitation measuring vessels, he constructed isograms (lines with the same height of precipitation) and assessed the uniformity visually as: very good, good, satisfactory or bad. The review of the literature shows that in practice, several methods are used to evaluate the quality of work of irrigation machines, e.g., coefficient of uniformity, coefficient of non-uniformity, degree of uniformity and coefficient of variation [13]. The uniformity of the application of the irrigation water with the irrigators is one of the important aspects of the indicators of the quality of the work of the irrigators. Research in the field of evaluating the quality of work of belt and wide-range irrigators points to the need to solve the problem. The quality of work of various machines showed the results of *CU* (irrigation uniformity coefficients according to Christiansen) and *CUH* (irrigation uniformity coefficient according to Heermann and Hein) in significant ranges from 60% to 92% depending on the research site [3,13,14]. To ensure optimal humidity conditions over the entire irrigated area, a high irrigation uniformity must be achieved, which is one of the most important indicators of irrigation quality [9]. In addition to proper maintenance, improving the quality of work in irrigation technology can also be achieved by designing and constructing new applicators [15]. For end users (farmers) in recent years, the quality of work and the possibility of remote control are the most important parameters which are considered. Due to the nature of the equipment, wide-area belt irrigators or micro-sprinklers, we require minimum values of quality of work evaluated by coefficients or degrees of uniformity [3,8,13,16]. Many authors state that wind speed is one of the main influencing factors in the quality of irrigation technology [5–7]. Recently, [17] found that there is typically 100% uniformity in the lateral direction of movement when there is constant movement of the pivot. It was also suggested that the research should be further examined if the overall runoff in such a method was used, and which sprinkler pattern has the most potential runoff that is connected to the intermittency movement of the center pivot. The coefficient of irrigation uniformity of irrigation equipment has a direct effect on the efficiency of application and crop yield [7,18–20]. Deployment of irrigation technology working on the principle of irrigation by sprinklers is an effective technology in compliance with technical and operational parameters and ensures further increases in productivity, decreases in manual labor and increased water savings, water protection and crop quality improvement [15,21].

The results of previous research have provided the basis for further methodological procedures for obtaining the results of the dependence of the evaluation of the quality of work of various irrigators of one species in variable input conditions. As there are several methods for evaluating the quality of work of wide-range irrigators, the aim of this study was to explore the possibilities of their application and evaluation by monitoring the coefficients of uniformity and non-uniformity. There were established two hypotheses,

where in the first it was assumed that the quality of work is given by the location in which the research is carried out (depends on the machine and conditions), and the second hypothesis to determine if the application of any parameter (uniformity, non-uniformity of new established) will indicate the differences in the quality of irrigation equipment.

## 2. Materials and Methods

The review of the literature shows that the issue of irrigation will need to be given increased attention, not only in terms of total water consumption, but also in terms of its precise distribution. Since the issue of quality assessment of work is dealt with at our workplace (Department of Machines and Production Biosystems/Faculty of Engineering/Slovak University of Agriculture in Nitra), it was decided to expand our knowledge in this area.

### 2.1. Experimental Area

Irrigation water was applied to crops such as potatoes, corn and sugar beets. Figure 1 presents areas of interest, land falling under the areas of agricultural enterprise SLOV-MARKT and Agrobiop, a joint stock company. The irrigation water represents the amount of water delivered per unit area in one uninterrupted time interval until the effective watering depth is reached. The soils located in this area are on the so-called "Trnava board", which has the shape of an almost isosceles triangle, where eroded loess of black soils predominates, which are mostly medium-heavy, deep and skeletal soils. Brown soils are almost non-existent in this area. Enterprises farm on a total area of about 2000 ha, of which 530 ha are irrigated. Irrigation is carried out with various irrigators such as wide-area (approximately 320 ha) and belt (140 ha) irrigators and economical irrigation technology (drip irrigation—in 2019, approximately 70 ha) was introduced. In 2020, drip irrigation was not carried out due to excessive costs. In most irrigated localities, it is possible to use wide-area irrigators, but on land where there are electrical networks or other obstacles, belt irrigators must be used. The importance of the application of the mentioned wide-range irrigators (pivot) mainly regards the time saving of the manual labor needs for the given irrigation work. In the case of the application of belt irrigators, it mainly regards smaller irrigation areas and greater time requirements of manpower.

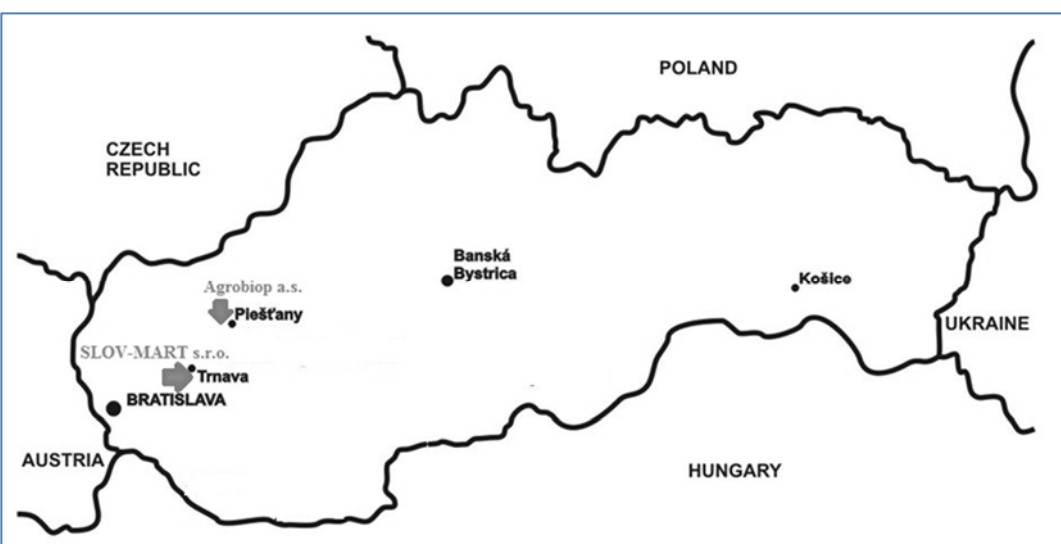

**Figure 1.** Location of measurement plain on western region on Slovak Republic map.

### 2.2. Applied Irrigation Machinery

In our portfolio of interest, there were wide-area irrigators of the two mentioned companies, which consisted of two pivot and two linear types. The first of the investigated irrigators has a total length of 540 m with a spray of 30 m with a long-distance sprayer.

For T-L sprinklers (T-L, Hasting, NE, United States), suspension hoses were used, at the end of which a Nelson regulator (pressure 0.1 MPa) was fitted, which was connected to a sprayer (Nelson R3000) and fitted with a prescribed type of nozzle from Nelson (# 10 to # 50, Table 1, manufactured by Nelson Irrigation Co., Walla Walla, WA, USA). The T-L linear sprinkler with a construction length of 280 m is also equipped with a long-jet sprayer with the possibility of extending the irrigation belt by up to 20 m. The number and fit of the nozzles are shown in Table 1 (nozzles ranging from # 10 to # 50, Nelson, Walla Walla, WA, USA).

**Table 1.** The nozzles on the investigated irrigators.

| *Ef* | | Irrigation Equipment Concept—Number and Type of Nozzles | | | | | | | | | | | |
|------|------|------|------|------|------|------|------|------|------|------|------|------|------|
| **NoS** | | **1** | **2** | **3** | **4** | **5** | **6** | **7** | **8** | **9** | **10** | **11** | **12** |
| F1 | Nd | 16 | 19 | 19 | 19 | 19 | 19 | 19 | 19 | 19 | 9 + gun * | - | - |
| | D | 10 ÷ 14 | 14 ÷ 19 | 20 ÷ 24 | 24 ÷ 28 | 28 ÷ 32 | 32 ÷ 35 | 35 ÷ 38 | 38 ÷ 40 | 40 ÷ 42 | 42 ÷ 50 | - | - |
| F2 | Nd | 3 | 7 | 9 | 7 | 9 | 7 | 9 | 7 | 9 | 7 | 9 | 3 + gun * |
| | Dn | 2.5 | 3.0 | 3.5 | 4.0 | 4.5 | 5.0 | 5.5 | 6.0 | 6.5 | 7.0 | 7.5 | 8.0 |
| F3 | Nd | 17 | 19 | 19 | 19 | 3 + gun * | - | - | - | - | - | - | - |
| | D | 17 | 17 | 17 | 17 | 17 | - | - | - | - | - | - | - |
| F4 | Nd | 17 | 20 | 19 | 18 | 8 + gun * | - | - | - | - | - | - | - |
| | D | 10 ÷ 36 | 36 | 36 | 36 | 36 ÷ 50 | - | - | - | - | - | - | - |

*Ef*—experimental field, NoS—number of spans, Nd—number of sprinklers for span, Dn—diameter of nozzle without gun, * end span of irrigation machine.

The source of irrigation water was a river or lake, and water was supplied to the irrigators through the main irrigation water distribution facility, which consisted of a pumping station (pumps, electric motors, filtration and control room), a pipe network and an end-point hydrant. The technical parameters of the inlet irrigators and the weather conditions for specific measurements are given in Table 2. Selected plots were irrigated with wide-range irrigation technology, linear and pivot irrigators, which were equipped with new or refurbished irrigation water distributors (Figure 2).

**Table 2.** The irrigation machine technical data and other details.

| *Ef* | | Technical Data | | | | | Weather Conditions | | | |
|------|------|------|------|------|------|------|------|------|------|------|
| | **Type** | *L*, m | *NoSs* | *NoSp* | *Ir*, mm | Ip, MPa | Wind Speed, m·s$^{-1}$ | | | Te, °C |
| | | | | | | | **T1** | **T2** | **T3** | |
| F1 | **T-L Pivot** | 540 | 9 | 177 | 16 | 0.7 | 3.7 | 1.3 | 0.3 | 28 ÷ 33 |
| F2 | **Fregat Koma Pivot** | 308 | 11 | 87 | 41 | - | 1.5 | 1.0 | x | 24 ÷ 28 |
| F3 | **Bauer Linear** | 230 | 4 | 77 | 32 | 0.6 | 2.4 | 0.2 | x | 30 ÷ 34 |
| F4 | **T-L Linear** | 280 | 4 | 82 | 13 | 0.6 | 3.2 | 2.8 | 0.3 | 34 ÷ 36 |

*Ef*—experimental field, *L*—length, *NoSs*—number of spans, *NoSp*—number of sprinklers, *Ir*—irrigation rate, Ip—input pressure, F1—speed setting 7 (T1, T2), speed setting 8 (T3), F2—undefined Ip and speed setting, F3—speed setting 6, F4—speed setting 6, Te—temperature.

One of the pivot irrigators was a Fregat DMU A308 (refurbished by Agref, Komárno, Slovakia), originally from a non-existent primary company (Fregat, Russia) and of an older type after a complete reconstruction with sprayers from Agref (Agref, Komárno, Slovakia, Figure 3). The sprayers were therefore manufactured on their own (cooperation with Agref), and the diameters of the spraying holes (range from 2.5 to 8.5 mm) were based on the experience of irrigators and monitoring of flows (not part of this study). The total length of the irrigator was 362 m with the possibility of spraying 15 m with a long-jet sprayer.

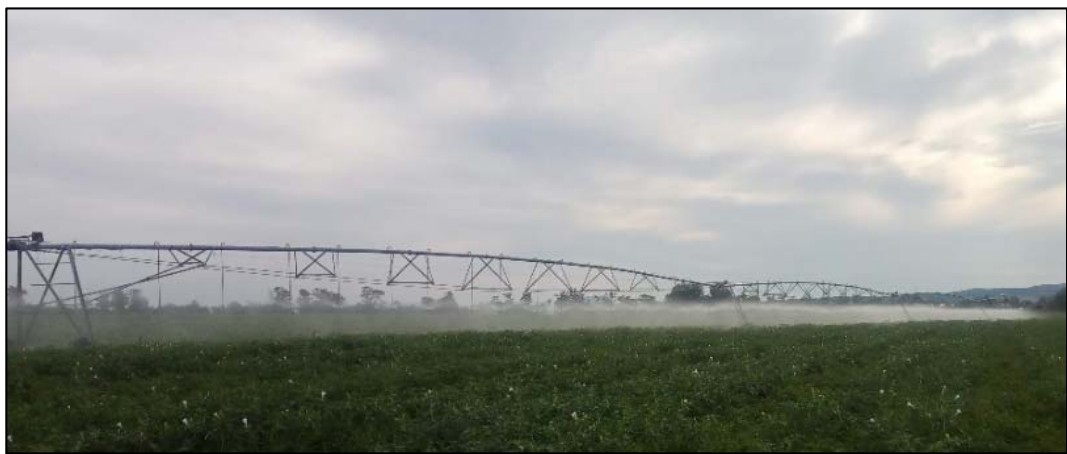

**Figure 2.** Example of used irrigation design used in experimental measurement which is depicted in practical conditions.

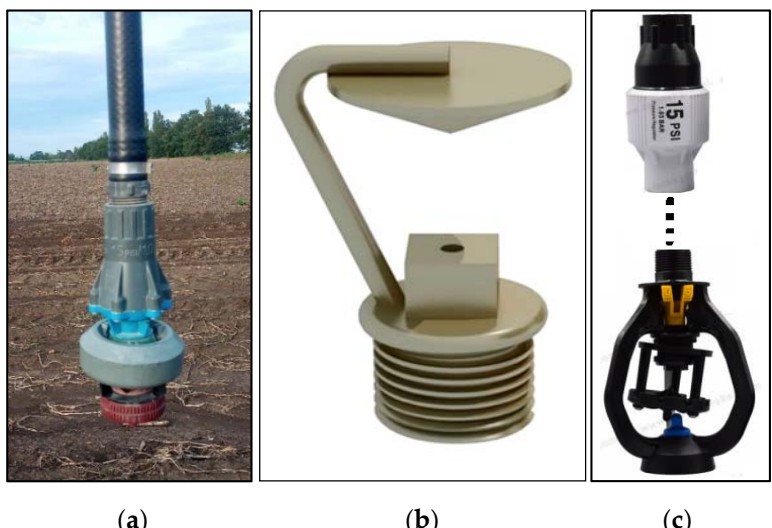

| (**a**) | (**b**) | (**c**) |

**Figure 3.** Sprayers on sprinklers, (**a**) T-L Pivot, linear—Nelson, (**b**) Fregat—Agref, (**c**) Bauer—I-Wob.

The last irrigator examined was the Bauer Linear with a total length of 230 m and the possibility of spraying 20 m with a long-jet sprayer. Another type of sprinkler, Senninger I-Wob (Senninger Irrigation Co., Clermont, FL, USA, Figure 3), was mounted on the sprinkler with a Senninger 15 PSI (0.1 MPa) pressure regulator.

*2.3. Quality of Work*

As can be seen from the available resources, standards and methodologies, there are several evaluation options for determining the quality of work, which are coefficients and degrees of uniformity or non-uniformity, and coefficient of variation. The practical evaluation of the quality of work was carried out according to certain principles of the ASAE standard [22], and the uniformity test was performed using rain gauge vessels to evaluate the uniformity of spraying. The rain gauge vessels had a diameter of 115 mm and a total height with a stand of 1000 mm above the ground. However, to evaluate the obtained results, it was also decided to apply other methods (newly established) than those mentioned by some authors [9,11,23].

To evaluate the quality of work, the first methodology of calculations according to Oehler was used [24], which is the calculation of the value of the average deviation $A$ (average error of individual precipitation heights from their arithmetic mean). From the achieved values of $A$, the value of non-uniformity $a$ is determined and according to the

following formula (for this comparison with other results in percent, the relationship was adjusted with a multiple of the constant 100):

$$a = \frac{A}{N_m} \cdot 100, \ \% \tag{1}$$

$N_m$ mean value, mm,
$A$ mean deviation, mm.

Since it was decided for the purpose of the study to divide all evaluations into two large groups (categories), namely the coefficients of uniformity and the coefficients of non-uniformity, a relation for the uniformity of irrigation $a_r$ was proposed and calculated according to the following relation (2):

$$a_r = 100 \cdot \left(1 - \frac{A}{N_m}\right), \ \% \tag{2}$$

The second method in the uniformity category, which was used to evaluate the results we obtained, is based on the frequently used and well-known method according to Heermann and Hein (*CUH*) [22,25]. The calculation is performed according to the following relations Equations (3) and (4):

$$CUH = 100 \cdot \left[1 - \frac{\sum_{i=m}^{n} \left\{ S_i \cdot \left| V_i - \overline{V} \right| \right\}}{\sum_{i=m}^{n} V_i \cdot S_i}\right], \ \% \tag{3}$$

$$\overline{V} = \frac{\sum_{i=m}^{n} V_i \cdot S_i}{\sum_{i=m}^{n} S_i}, \ \text{mm} \tag{4}$$

$n$ number of precipitation measuring vessels,
$i$ a number intended to identify a specific rain gauge vessel beginning with $i = 1$ for the vessel closest to the beginning and ending with $i = n$ for the rain gauge vessel furthest from the pivot,
$V_i$ irrigation water in the $i$-th rain gauge vessel, mm,
$S_i$ the distance of the $i$-th rain gauge vessel from the pivot, m,
$\overline{V}$ mean irrigation water, mm,
$\left| V_i - \overline{V_i} \right|$ absolute value of deviations from the average irrigation water, mm.

The third method, according to Christiansen (*CU*) [26], in the category of determination of uniformity, uses a simpler calculation and is most common for belt irrigators. The relationship for the calculation is as follows (5):

$$CU = 100 \cdot \left[1 - \frac{\sum_{i=1}^{n} \left| V_i - \overline{V} \right|}{n \cdot \overline{V}}\right], \ \% \tag{5}$$

$V_i$ irrigation water in the $i$-th rain gauge vessel, mm,
$\overline{V}$ average irrigation water, mm,
$n$ number of rain gauges, i.e., the number of elementary areas into which the area is divided.

The fourth method, according to Wilcox and Swailes [27], is aimed at calculating the irrigation coefficient $C_{ws}$ from the ratio of the standard deviation and the average irrigation water according to the following relationship (6):

$$C_{ws} = 100 \cdot \left[1 - \frac{\sigma}{\overline{V}}\right], \ \% \tag{6}$$

$\sigma$ standard deviation, mm,
$\overline{V}$. mean irrigation dose, mm.

The fifth method in the category of irrigation uniformity was the procedure according to Voigt [28] and is based on the calculation of the degree of non-uniformity $\gamma$ (Equation (11)), which also considers the distance of the measured points of the sprayer $r_i$. To be able to compare the obtained results with other calculations of uniformity coefficients, it was decided to introduce a factor $\gamma_r$, which is calculated as follows:

$$\gamma_r = 100 \cdot \left( 1 - \frac{\sum_{i=0}^{n} r_i \cdot |h_m - h_i|}{h_m \cdot \sum_{i=0}^{n} r_i} \right) \tag{7}$$

$\gamma_r$ degree of uniformity, %,
$h_m$ average amount of precipitation on the examined area, mm,
$h_i$ amount of precipitation on elementary surfaces, mm,
$r_i$ distance of the measured vessels from the sprayer, m.

Another method is to calculate the value of the irrigation uniformity coefficient according to Hart and Reynolds [29], who proposed the "distribution efficiency", i.e., the value based on numerical integrations of the normal distribution function, while the distribution efficiency is determined by first selecting the target $CU_{hr}$ (in the range of 70 to 100%) and the target "adequately irrigated area" (in the range of 50 to 100%) (Equation (8)). The calculation is as follows:

$$CU_{hr} = 100 \cdot \left[ 1 - \frac{0.798 \cdot \sigma}{\overline{V}} \right], \% \tag{8}$$

$\sigma$ standard deviation, mm,
$\overline{V}$ mean irrigation water, mm.

The seventh parameter for evaluating the quality of irrigation technology is the coefficient according to Criddle et al. $CU_{cr}$ [30], in the equation limiting the deviations of the averages to quarterly irrigation water (Equation (9)).

$$CU_{cr} = 100 \cdot \left[ 1 - \frac{\sum_{i=1}^{n/4} |V_i - \overline{V}|}{(n/4) \cdot \overline{V}} \right], \% \tag{9}$$

$V_i$ irrigation water in the $i$-th rain gauge vessel, mm,
$\overline{V}$ average irrigation water, mm,
$n$ number of containers, i.e., the number of elementary areas into which the area is divided.

The last parameter for evaluating the quality of work was the coefficient of irrigation uniformity according to Beale and Howell $CU_{bh}$ [31], where the authors used the concept of average deviations and limited the equation to the highest quarter of irrigation water. The equation was proposed as follows:

$$CU_{bh} = 100 \cdot \left[ 1 - \frac{\sum_{i=\frac{3}{4n}+1}^{n} |V_i - \overline{V}|}{(n/4) \cdot \overline{V}} \right], \% \tag{10}$$

$V_i$ irrigation water in the $i$-th rain gauge vessel, mm,
$\overline{V}$ average irrigation water, mm,
$n$ number of containers, i.e., the number of elementary areas into which the area is divided.

The degree of non-uniformity $\gamma$ (the second method for determining the non-uniformity, the first was the coefficient $a$, Equation (1)) is calculated according to the following relation (for our comparisons, the value was converted to % of expression):

$$\gamma = \frac{\sum_{i=0}^{n} r_i \cdot |h_m - h_i|}{h_m \cdot \sum_{i=0}^{n} r_i} \cdot 100, \% \tag{11}$$

$\gamma$ degree of non-uniformity, %,

The third method for determining the non-uniformity (the first method—calculation *a*) was the calculation of the coefficient of variation $C_v$ [32], which represents the dependence of the standard deviation $\sigma$ and the average irrigation water $h_m$ (the value according to our findings is related to a comparison, we changed the value to % expression):

$$C_v = \frac{\sigma}{h_m} \cdot 100, \tag{12}$$

Another method was based on the procedures of Hofmeister (1961), where the output is the degree of non-uniformity $E_f$ (the value is related to the value of the uniformity coefficient *CU* according to Christiansen and, for our comparisons, the values were converted to % expression):

$$E_f = \frac{\sum_{i=1}^{n} \left| h_i - \overline{h} \right|}{n \cdot \overline{h}} \cdot 100, \ \% \tag{13}$$

$E_f$ degree of non-uniformity, %,
$\overline{h}$ average amount of irrigation water on the examined area, mm,
$h_i$ amount of irrigation water on elementary surfaces, mm,
$n$ number of elementary areas.

### 2.4. Final Evaluation of Datasets

From the applied equations for the calculation of coefficients and degrees of uniformity and non-uniformity and coefficients of variation, it was assumed that the results would not be comparable or observable without statistical significance tests. That is why the obtained results are followed by demonstrable or unprovable changes in the monitoring of the influence of different uniformity coefficients and, on the other hand, also in the monitoring of different input conditions. For quality and sufficient evaluation of the results, it was necessary to use statistical analyses. Therefore, the statistical apparatus STATISTICA [33] was used, in which a one-factor ANOVA analysis was used to evaluate and compare the results of different coefficients of irrigation uniformity (Equation (14)), but also to evaluate the impact of different sites, e.g., field conditions (Equation (15)):

$$y_{ij} = \mu + C_i + e_{ij}, \ \text{mm} \tag{14}$$

$$y_{ij} = \mu + F_i + e_{ij}, \ \text{mm} \tag{15}$$

$y_{ij}$ measured value,
$\mu$ overall mean,
$C_i$ effect of the uniformity coefficient,
$F_i$ effect of the field conditions,
$e_{ij}$ random error with mean 0 and variance $\sigma 2$.

In view of the results obtained when the null hypothesis is rejected, the Tukey honestly significant difference (HSD) test was applied at a significance level of 95% and the assumed alternative hypothesis was that it is not possible to use different parameters for the wide-range irrigator category. As another test, the Duncan test was used to evaluate the results. To compare the results with respect to the rejected null hypothesis, the results obtained were tested by Dunnett's test, which is based on a comparison with a control group (in this study, it was *CUH* values). Finally, an overall evaluation of the achieved results was processed.

## 3. Results and Discussion

The research work was aimed at evaluating the quality of irrigation technology during the growing seasons of cultivated crops. The value of the irrigation water depended on the input conditions, such as the crop, the time of the experiments and the soil conditions. The achieved results monitored by rain gauge vessels were evaluated by twelve coefficients of uniformity and non-uniformity of irrigation (eight parameters of uniformity and four coefficients of non-uniformity). To be able to compare the results obtained with each

other, the values were converted into percentages. As uniformity coefficients were used to evaluate the results, we proposed to transform some evaluation methods from non-uniformity to uniformity ($a_r$, $\gamma_r$). The first measurements (T1), focused on irrigation uniformity, were performed on all four monitored irrigators, and the technical parameters and weather conditions are given in Table 2. The wind speed reached a range from 1.5 to 3.7 m·s$^{-1}$ depending on the location. A graphical representation of the results of the first round of testing is shown in Figure 4 for irrigation uniformity and Figure 5 for irrigation non-uniformity. The results show differences both regarding the monitored irrigators and regarding the applied coefficient of uniformity or non-uniformity of irrigation. The values for the irrigation uniformity coefficients ranged from 67.58 to 95.17%. The figure clearly shows the differences, where for the first seven parameters the results at the level of significance of 95% did not show significant differences ($p = 0.69$, $p > 0.05$, F < Fcrit). However, statistically significant differences ($p = 0.03$, $p < 0.05$, F > Fcrit) were demonstrated when the last irrigation uniformity coefficient was introduced (all eight irrigation uniformity coefficients). This means that with the introduction of all eight coefficients, there is a statistically significant difference in the values of the uniformity coefficients between at least the two evaluation methods. If the null hypothesis is rejected, an alternative hypothesis emerges. This means that not all mean values are the same (at least one of the mean values is different from the others). If the analysis of variance rejects the global null hypothesis about the effect of a factor (selection of the irrigation uniformity coefficient), the analysis needs to be supplemented by other methods for evaluating existing differences. These multi-comparative tests (tests for multiple comparisons) subsequently give the results the statistical significance of the individual differences in the mean values for all possible pairs of the compared groups. From the most used tests for multiple comparisons of all pairs of groups with each other in the experiment, the Tukey test was chosen. The results according to the Tukey test, based on the obtained and achieved average values, show at the level of significance ($\alpha = 0.05$) for the division of evaluation methods (irrigation uniformity coefficients) into two groups to exclude from the eight evaluation methods the lowest or highest value of the average parameter ($C_{ws}$ or $CU_{bh}$, $p = 0.69$, or $p = 0.11$). This fact is also pointed out by the graphical representation of the obtained results (Figure 4), where the average value of $C_{ws}$ is the lowest and $CU_{bh}$ the highest in all observations. This means that if we exclude one of the above coefficients from the evaluation, based on the analysis of variance at the significance level $\alpha = 0.05$, it can be said that different coefficients of irrigation uniformity do not have a statistically significant effect on its selection. Additionally, using the Duncan test, the results show that the first group of data consists of irrigation uniformity coefficients without the $CU_{bh}$ coefficient at the level of significance ($\alpha = 0.05$). In the case where one of the groups in the experiment serves as a control group (e.g., CUH), the mean values of the other experimental groups were compared with respect to the control (Dunnett's test). Differences were found to be statistically significant for the coefficients $C_{ws}$ (Sig. = 0.51) and $CU_{bh}$ (Sig. = 0.06).

When monitoring the influence of different input parameters (different field conditions), it was found, based on the achieved results and analysis of variance at the significance level $\alpha = 0.05$, that different input conditions have a statistically significant effect on the selection ($p = 0.0011$, $p < 0.05$, F > Fcrit). It can be stated from the results that in no case can different input conditions be confused, in other words, results in one location cannot be determined based on results in another.

As follows from the methodological procedures, four different parameters of irrigation non-uniformity were also used to evaluate the quality of work. The results obtained and calculated for these parameters in the first round of testing (T1) ranged from 13.02 to 32.42%. When monitoring the coefficients of non-uniformity (Figure 5), significant differences were observed in one wide-area irrigator (designation F4). Statistical evaluation of the results by one-way ANOVA (dependence of the difference of coefficients) on the level of significance of 95% did not show significant differences ($p = 0.33$, $p > 0.05$, F < Fcrit). However, when

examining the dependence on the input field conditions, a statistically significant effect was demonstrated ($p = 0.0007$, $p < 0.05$, F > Fcrit).

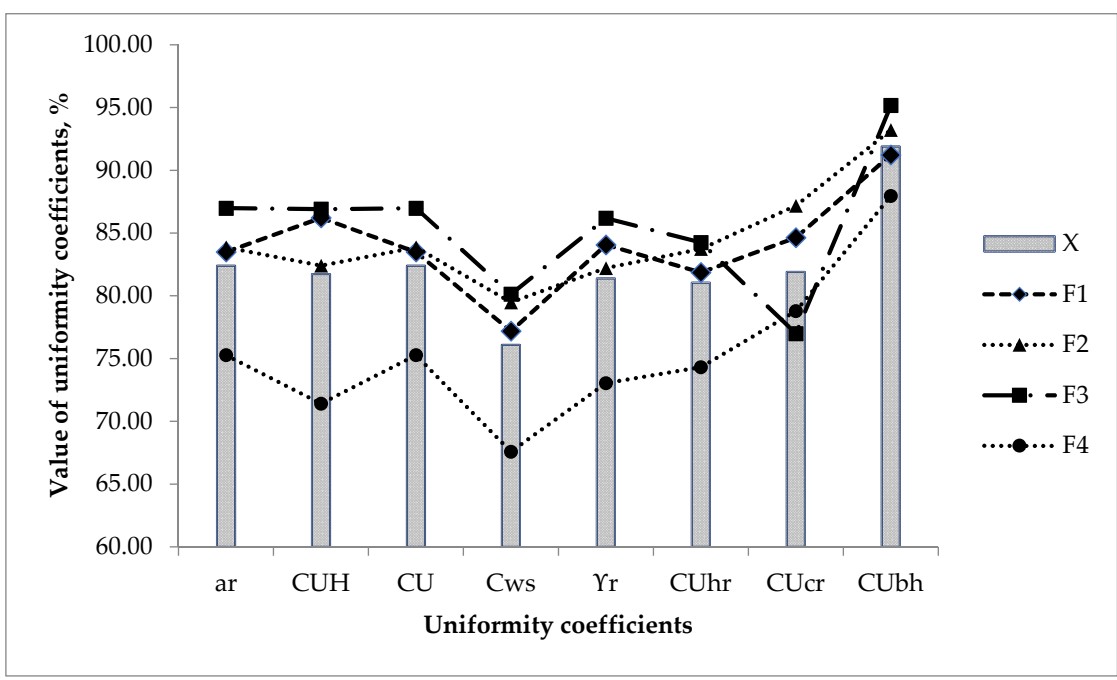

**Figure 4.** The results of different coefficients of uniformity for all experimental fields, for T1 where: ar—coefficient of uniformity according to Oehler, CUH—coefficient of uniformity according to Heerman and Hein, CU—coefficient of uniformity according to Christiansen, Cws—coefficient of uniformity according to Wilcox and Swailes, γr—coefficient of uniformity according to Voigt, CUhr—coefficient of uniformity according to Hart and Raynolds, CUcr—coefficient of uniformity according to Criddle et al., X—average values in individual samples.

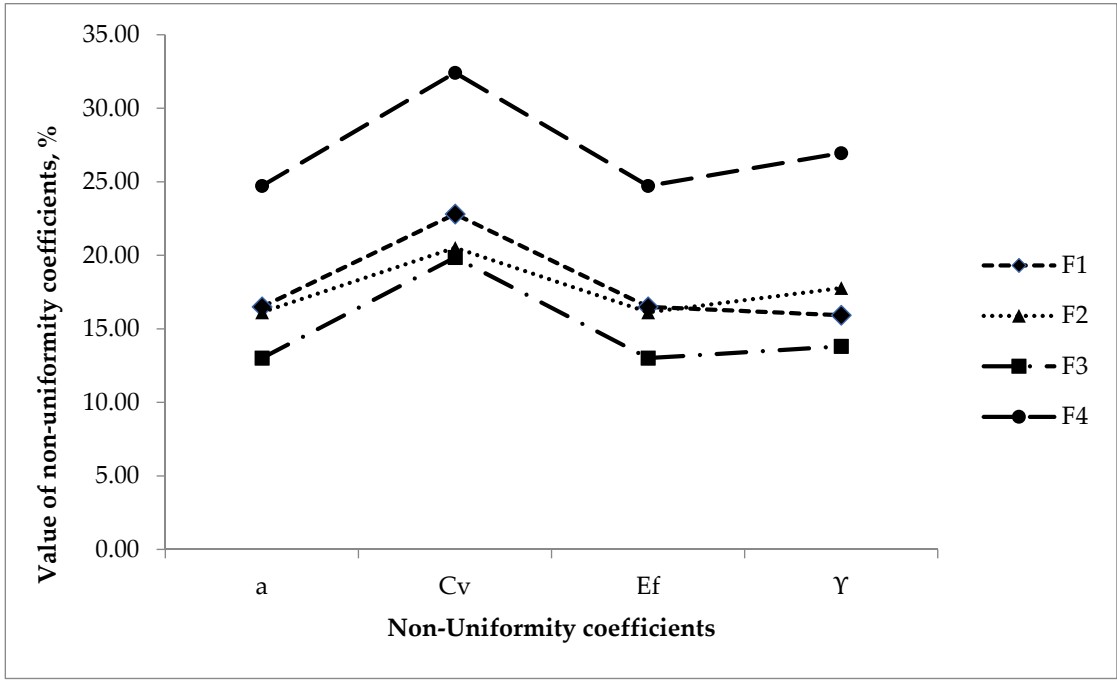

**Figure 5.** The results of different coefficients of non-uniformity for all experimental fields, for T1 where: a—coefficient of non-uniformity according to Oehler, Cv—coefficient of variation, Ef—degree of non-uniformity according to Hofmeister, γ—coefficient of non-uniformity according to Voigt.

In view of the results obtained and some lower values of the irrigation uniformity coefficients achieved during the first test (T1), we proposed a second test (T2) under the control of input conditions on the equipment and weather conditions. In all four input conditions, the wind speed decreased, the total range was from 0.2 to 2.8 m·s$^{-1}$. Additionally, in this round of testing, we applied eight coefficients of irrigation uniformity (Figure 6) and four coefficients of non-uniformity (Figure 7) to evaluate the results. The values of the irrigation uniformity coefficients ranged from 71.7 to 95.88%. Differences were monitored in both cases (i.e., the dependence on the parameter factor, and the dependence on the input conditions), where when examining the statistical dependence on the choice of the uniformity coefficient, there was no significant effect ($p = 0.31$, $p > 0.05$, F < Fcrit). Compared to the first round of testing, the results of statistical significance have changed, which means that with a sufficiently balanced overall quality of work, any of the coefficients can be applied. Since the established null hypothesis of the use of different coefficients of irrigation uniformity has been confirmed, no further tests need be performed. However, the achieved results of the effect of various field conditions showed, based on the analysis of variance at the level of significance $\alpha = 0.05$, a statistically significant effect on their selection ($p < 0.05$, F > Fcrit). These results are also confirmed Figure 7. The fourth irrigator (F4) showed the lowest quality of work, which could be caused by the highest value of wind speed during the test of irrigators in the second round (T2). When evaluating the coefficients of irrigation non-uniformity, the null hypothesis was confirmed ($p = 0.061$, $p > 0.05$, F < Fcrit), i.e., any of the above coefficients can be applied. However, in the second round (T2) of testing, the result, which was also achieved in the first testing (T1), was confirmed by a statistically significant difference with respect to different input conditions ($p < 0.05$).

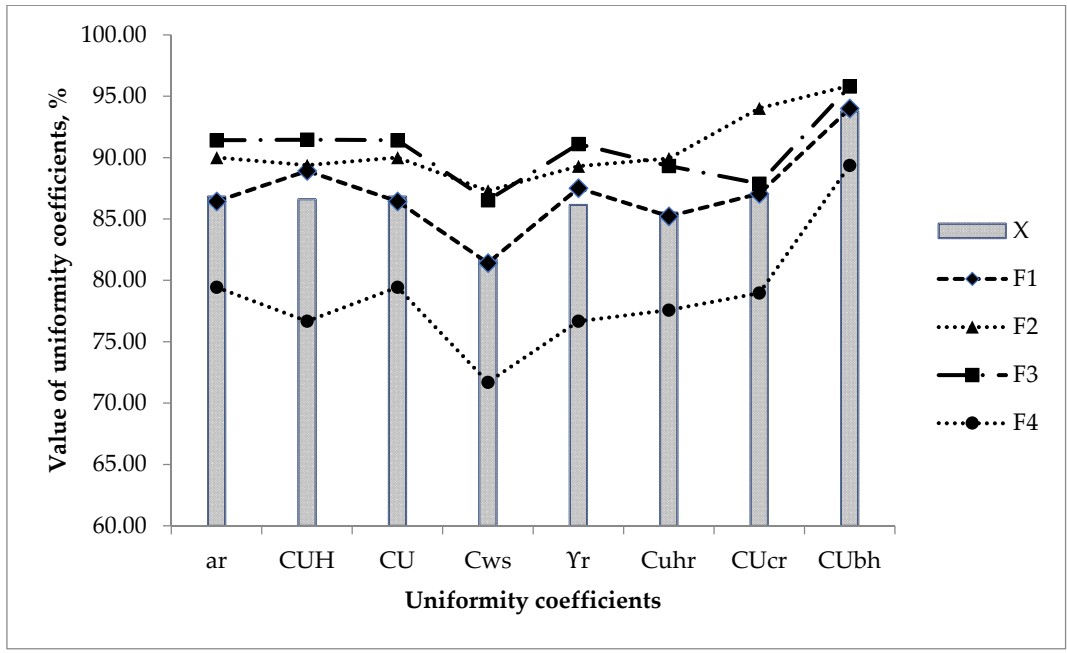

**Figure 6.** The results of different uniformity coefficients for all experimental fields, for T2 where: ar—coefficient of uniformity according to Oehler, CUH—coefficient of uniformity according to Heerman and Hein, CU—coefficient of uniformity according to Christiansen, Cws—coefficient of uniformity according to Wilcox and Swailes, γr—coefficient of uniformity according to Voigt, CUhr—coefficient of uniformity according to Hart and Raynolds, CUcr—coefficient of uniformity according to Criddle et al., X—average values in individual samples.

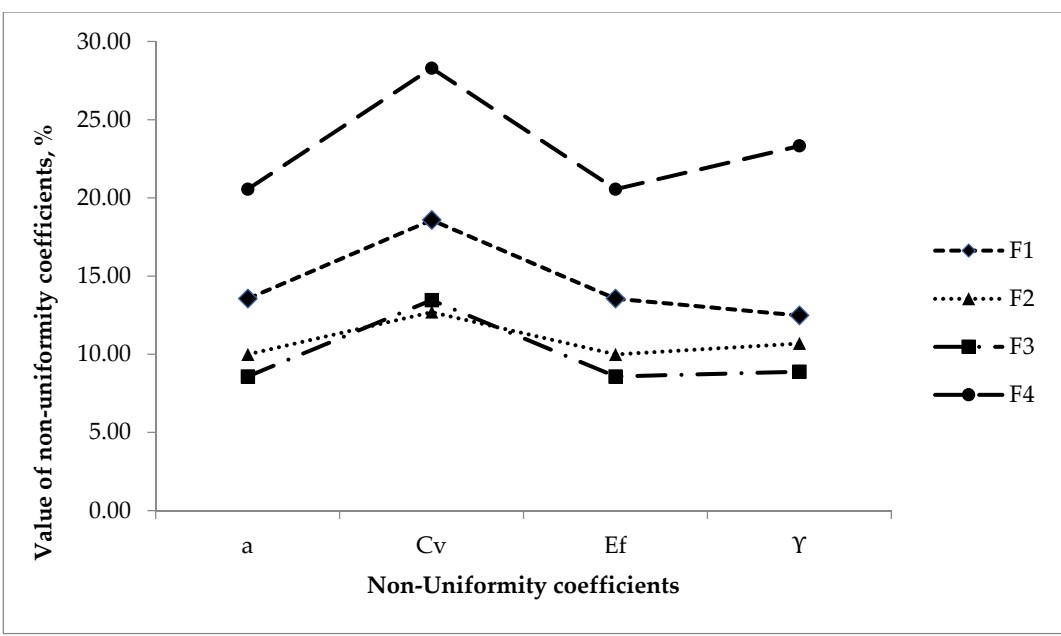

**Figure 7.** The results of different coefficients of non-uniformity for all experimental fields, for T2 where: a—coefficient of non-uniformity according to Oehler, Cv—coefficient of variation, Ef—degree of non-uniformity according to Hofmeister, γ—coefficient of non-uniformity according to Voigt.

Based on the results obtained in the second round (T2) of testing, a third round (T3, in Table 2) was deployed, where a minimal effect of weather conditions was observed. A graphical representation of the results, which presents the results obtained in all rounds of testing for irrigators (T1, T2 and T3) under field conditions (F1 and F4), is shown in Figure 8 (irrigation uniformity) and Figure 9 (irrigation non-uniformity). Coefficients of irrigation uniformity ranged from 75.97 to 92.16% (only for the third round of testing). The values of irrigation non-uniformity reached values from 14.45 to 24.03% (only for the third test). Tests of one-way ANOVA were also performed at the level of reliability ($\alpha = 0.05$), to evaluate the achieved results of all tests for field conditions F1 and F4. The results showed that different coefficients of irrigation uniformity have a statistically significant effect on its selection ($p < 0.05$, F > Fcrit). Therefore, a step was taken as in the first testing, namely, to eliminate the last irrigation uniformity coefficient $CU_{bh}$. The results subsequently confirmed the null hypothesis ($p = 0.13$, F < Fcrit). For the results of irrigation non-uniformity, shown in Figure 9, when comparing two different field measurements with three replicates, a positive result of reducing the values and thus increasing the quality of work ($p > 0.05$, when testing for the applicability of the coefficient) was achieved.

Overall, the results indicate that evaluations by different irrigation uniformity coefficients according to Oehler, Heermann and Hein, Christiansen, Wilcox and Swailes, Voigt, Hart and Reynolds, Criddle et al. and Beal and Howell obtained different results, which were then statistically evaluated. However, some of the values obtained are very close and the choice between them would not unduly affect the overall result. From three different experiments under two different conditions, the effects of wind speed on the quality of irrigation technology were also shown. It is clear from the results that the quality of work increased as the effect of wind speed was reduced. The largest deviations occurred in the Beal and Howell coefficient (with irrigation uniformity) and the coefficient of variation (with irrigation non-uniformity). The change in and influence of different inlet conditions essentially depended on the correct deployment of the irrigation nozzles and the correct functionality (and maintenance) of the entire machine. The value of irrigation uniformity did not exceed 100% in any case.

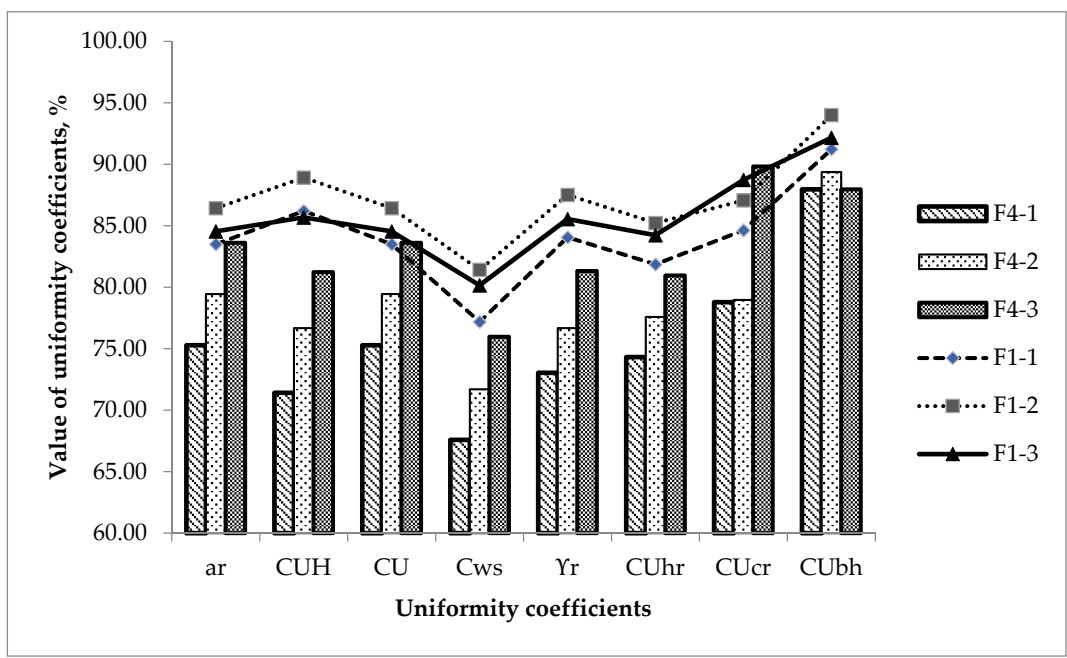

**Figure 8.** The results of different coefficients of uniformity for all experimental fields, all for F1 and F4 where: ar—coefficient of uniformity according to Oehler, CUH—coefficient of uniformity according to Heerman and Hein, CU—coefficient of uniformity according to Christiansen, Cws—coefficient of uniformity according to Wilcox and Swailes, γr—coefficient of uniformity according to Voigt, CUhr—coefficient of uniformity according to Hart and Raynolds, CUcr—coefficient of uniformity according to Criddle et al.

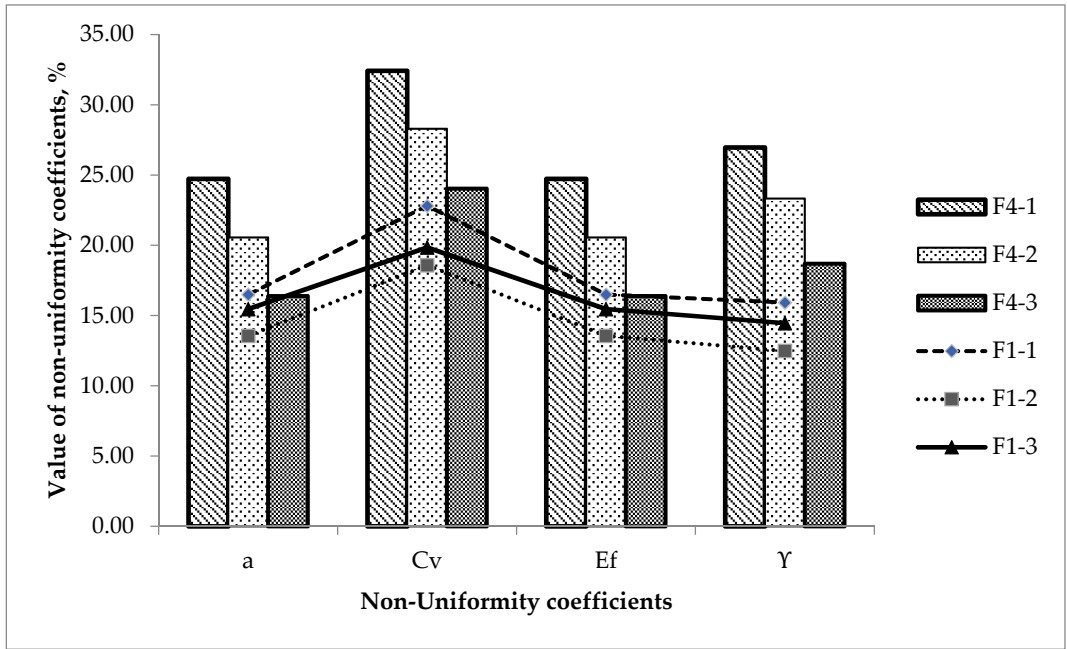

**Figure 9.** The results of different coefficients of non-uniformity for all experimental fields, all for F1 and F4 where: a—coefficient of non-uniformity according to Oehler, Cv—coefficient of variation, Ef—degree of non-uniformity according to Hofmeister, γ—coefficient of non-uniformity according to Voigt.

The available sources of several research works show that the quality of work, i.e., the coefficient of irrigation uniformity, is given mainly by weather conditions (uncontrollable wind direction and speed), but also by design variables of the system (irrigator type, nozzle size and type, irrigation water pressure and spacing between nozzles) [34]. In terms of methods and procedures for monitoring the coefficients of irrigation uniformity, it is about

expressing the uniformity of irrigation water distribution for different irrigation systems. In the last few decades, more of them have been proposed and many authors have addressed the issue of their follow-up ([3,7,8,15,32] and other authors). One of the most well-known irrigation uniformity coefficients, which has been put into practice in terms of quality of work, is the Christiansen uniformity coefficient. The results of applying different irrigation uniformity coefficients were also applied in the province of Kurdistan where the researchers emphasized the fact that different uniformity coefficients depend on field conditions and uniformity coefficients should not be used for any other field conditions [23].

The deployment of precision agriculture, in particular the precision irrigation sector, is a modern approach that uses information technology to manage the whole farm (such as GNSS satellite positioning data, remote sensing and proximal data collection). In 2011, [35] emphasized that this is beneficial, and in a comprehensive study, five irrigation performance indicators were used to assess the various combinations of tillage and traffic, namely, the Wilcox–Swailes coefficient of uniformity; application efficiency; deep percolation ratio; tail water ratio; and adequacy.

Modern technologies based on this principle will make it possible to achieve a reduction in costs and environmental impacts and to optimize the return on inputs, essentially in conjunction with environmental impacts of rain fall experimentally proved by artificial irrigation as simulated by [36] and further expanded by [37]. There were also field experiments at selected localities, where the uniformity of irrigation water distribution and the quality of irrigation technology work were determined. The correct quality of work and the correctly set amount of irrigation water (irrigation dose) is used to achieve a reduction in input costs with accurate irrigation in place [38]. This concept was also proved by [39] in experimental conditions. It was concluded that good uniformity gives more uniform production, but irregularity may water some places either more than enough or less than required, therefore it is considered as an economic loss in terms of quantity or work.

Further field experiments were carried out on the farm of the Faculty of Agriculture of Sebha University in Libya. The inlet working conditions involved a change in operating pressures and the height of the irrigators. During the research measurements, the quality of the work and its variability with respect to the change in input conditions were monitored. Collection vessels with a diameter of 120 mm and a height of 200 mm were used to collect the irrigation water. From the achieved measured irrigation water doses, they determined the coefficients of irrigation uniformity *CU* and uniformity of irrigation water distribution where they found that the investigated parameters increase with increasing pressure. The results showed that the average of the maximum *CU* values reached 91.37%, and the average of the minimum *CU* values was 78.21% [40].

In the case of the presented works and the researched methods and their mutual comparison, the authors did not include some coefficients of uniformity in the evaluation and did not use the coefficients of non-uniformity for evaluation. That is why the coefficients of irrigation non-uniformity and the proposed selected coefficients of uniformity, which were derived from other methods of evaluation, were applied in our evaluations.

One-way ANOVA was used for statistical evaluation of the results. Each of the possible tests available for multiple comparisons has slightly different properties, differing mainly in how they treat the error size of the first type $\alpha$ (the level of significance of the test) during testing. In our research, the Tukey "honestly" significant difference (HSD) test methodology was used in our research.

The characteristics of the selected four irrigators in different conditions of use demonstrably showed statistically significant differences. The results were monitored by means of experiments where rain gauge vessels were distributed along the entire irrigator, which consisted of a predefined number of sprinklers. Irrigation water in individual rain gauge vessels was evaluated by two main groups of parameters, namely the parameters of irrigation uniformity on the one hand and the parameters of non-uniformity on the other hand. The results were evaluated by a statistical one-factor ANOVA, and subsequently the results were subjected to another test based on the evaluation of hypotheses and the

assumed alternative hypothesis. A test was applied to compare the mean values of the subgroups of datasets with the averages of the other subgroups, adjusting the confidence level for each interval, followed by a test to estimate specific differences between pairs of means and finally a test to compare results based on a comparison with the control. In this case, the values of the coefficients were authoritatively determined according to Heermann and Hein *CUH*. The results also pointed to the possibility of using different coefficients of irrigation uniformity (out of the eight examined), but only with increased quality of work (i.e., more balanced irrigation uniformity and a minimum value of 81% for any examined uniformity parameter). New parameters for evaluating the results were introduced, which reflect the comparability of the results, and the effects of wind speed on the quality of work were confirmed. The results are of great importance for increasing the quality of sprinkler irrigation.

### 4. Conclusions

In this paper, attention has been paid to the evaluation of the quality of work of selected wide-range irrigators in various input conditions, where the first hypothesis was confirmed and the dependence of quality of work depends on the input conditions ($p < 0.05$). In the case of confirming the second established hypothesis, concerning the possibility of applying different coefficients of uniformity and non-uniformity, the results showed the possibility of using up to seven different coefficients of irrigation uniformity (out of the eight examined), but only with increased quality of work (more balanced irrigation uniformity, at least 81%). The difference in this study from the others was in the use of different input conditions under which new parameters (coefficients for the evaluation of uniformity or non-uniformity) were used to evaluate the results, which reflect the comparability of results and subsequently partially confirmed the effects of wind speed on work quality. The results provide knowledge for practical and research institutions. As part of further research, we propose a more detailed examination of the effect of wind (speed and direction), which would first be simulated in laboratory conditions on a proposed console for applying irrigation water with an additional device to change the wind speed (the device is still in the production process). Subsequent research should also focus on obtaining results from field research. Lastly, we would like to focus on the effect of the amount of irrigation water on the erosive effects in soil.

**Author Contributions:** Conceptualization, J.J. and K.K.; methodology, P.D.; validation, J.J., K.K. and P.D.; formal analysis, P.D. and J.M.; investigation, P.D.; resources, J.J.; data curation, P.D. and J.M.; writing—original draft preparation, J.J.; writing—review and editing, K.K.; visualization, J.J. and K.K.; supervision, J.J. and V.S.; project administration, V.S. and J.M.; funding acquisition, J.J. All authors have read and agreed to the published version of the manuscript.

**Funding:** This publication was supported by the Operational Programme Integrated Infrastructure within the project: Sustainable smart farming systems considering the future challenges 313011W112, financed by the European Regional Development Fund.

**Institutional Review Board Statement:** Not applicable.

**Informed Consent Statement:** Not applicable.

**Acknowledgments:** Many thanks are due to the editor and reviewers for their valuable comments to improve the quality of this paper.

**Conflicts of Interest:** The authors declare no conflict of interest.

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
