# Peer review of "Evaluation of the Quality of Irrigation Machinery by Monitoring Changes in the Coefficients of Uniformity and Non-Uniformity of Irrigation"

_agronomy, doi:10.3390/agronomy11081499_

Round 1

Reviewer 1 Report

Dear authors,

I congratulate you on a hard work, and interesting manuscript with useful results. Kindly, find some issues that should be addressed here below. General comments:

The introduction section could be improved in terms of English editing, and to give better insight into the study topic as well as the importance of the study. English editing refers to better choice of irrigation related terms. Further, the study goal is not clear enough. Kindly add sentence at the end of the introduction section that clearly states the study hypothesis. Also, the irrigation terms are poorly presented. For example, for amount of irrigation water it is stated the irrigation dose. I encourage you to check this in available irrigation glossaries. Conclusion sounds like abstract, kindly rewrite it. Give the insights in main study results and emphasize what is that your study makes different from others, what is the subject for further studies, are there any weak points of the study?

Specific comments:

Introduction section

L45 – „… most even and sufficient harvest …“   Maybe it sounds better „high yield and yield stability“

L47 – 48  Is this really true? Only by maintenance and inspection of nozzles and sprayers you could get high and stable yield?

L48-49 – Maybe it sounds better to say that Irrigation system efficiency is assessed by the correct intensity and uniformity of irrigation water distribution.

L50 – Check the reference style. Is it [7,9] or [7-9]

Reference 10 is missing

L59 – Kindly indicate what the CU and CUH stands for!

Material and method section

L87 – Kindly indicate what is the meaning of irrigation dose?

L91 – and further in manuscript, Kindly change “earth” to soil

L 93 – … 530 ha are irrigated.

L93, 94 – Kindly indicate the meaning of wide-area and belt irrigators

L109-112 – Is this sentence truly necessary?

L145-159 – Check if this section goes to introduction section instead of Material and methods

L221 – It seams that the reference 26 is missing

Results and discussion

L301-302 – Kindly, rewrite this sentence, it is not clear enough

L315 – Graphical representation could be changed to figure

Figure 4. uniformity coefficient can be presented from 60% (not zero) for better presentation

L356-358 – Is this conclusion?

L358 – maybe it would be better to use location (or something else) instead of company

Author Response

We are very grateful for reviewer valuable input. Please, find attached document for detailed replies.

Reviewer 2 Report

A BRIEF SUMMARY

The paper titled “Evaluation of the Quality of Irrigation Machinery by Monitoring Changes in the Coefficients of Uniformity and Inconsistency of Irrigation” presents a good topic for readers of this Journal. The topic represents a line of research as interesting as studied. The document needs to be better structured. English needs to be improved. However, results are poor described and analyzed in the paper. Finally, some question remain after reading the paper. Below is the list of some questions that need to be addressed.

  • Why the references is so few? Probably a more accurate references research could help to add value for this topic. I suggest to add some more references (see specific comments) especially in the "part 1 (introduction)" of the paper. I suggest the following paper:

Tauro, F., Grimaldi, S., Petroselli, A., Rulli, M. C., and Porfiri, M.: Fluorescent particle tracers in surface hydrology: a proof of concept in a semi-natural hillslope, Hydrol. Earth Syst. Sci., 16, 2973–2983,

  • I note that the paper lacks of a real conclusion section. I believe it would be appropriate to specify the novelties of this manuscript with respect previous studies. These novelties must also be reported in the conclusions.

  • Please, add more discussion on wind effect. Several recent study has highlighted this problem. For discussion, you can see:

Apollonio, C.; Petroselli, A.; Tauro, F.; Cecconi, M.; Biscarini, C.; Zarotti, C.; Grimaldi, S. Hillslope Erosion Mitigation: An Experimental Proof of a Nature-Based Solution. Sustainability 2021, 13, 6058. https://doi.org/10.3390/su13116058

  • The authors have to upgrade the “discussion” section, too poor in the current state. I suggest you, also a more large discussion of figures.

  • Finally, I hope to read in the revised version suggestions for future developments of this interesting work.

SPECIFIC COMMENTS

Figures 1 – 2 - 3: I suggest replacing figures. These are not ready for publication. By them, I cannot understand actual informations on experimental plot.

Conclusions: It seems an additional discussion. I suggest moving it in discussion and then rewriting conclusions, highlighting the novelty and future developments of this study.

Author Response

We are very grateful for reviewer valuable input. Please, find attached document for more detailed replies.

Round 2

Reviewer 2 Report

The paper has been improved. In my opinion it is ready for pubblication.